# Origin of heavy rare earth mineralization in South China

Cheng Xu[1], Jindřich Kynický[2,3], Martin P. Smith[4], Antonin Kopriva[3], Martin Brtnický[2,3], Tomas Urubek[3], Yueheng Yang[5], Zheng Zhao[6], Chen He[1] & Wenlei Song[1,2]

Heavy rare earth elements (HREE) are dominantly mined from the weathering crusts of granites in South China. Although weathering processes occur globally, no economic HREE resources of this type have yet been found outside China. Here, we report the occurrence of unidentified REE minerals in the granites from South Chinese deposits. They contain high levels of both HREE and light REE, but are strongly depleted in Ce, implying high oxidation state. These REE minerals show higher initial Nd isotope than primary REE-rich minerals ($\varepsilon$Nd($t$) = 0.9 ± 0.8 versus − 11.5 ± 0.5). The mineralized weathering crusts inherited REE signature of the granites, but show more Ce depletion and more overall concentration of the REE. We propose, therefore, that highly oxidized, REE-rich fluids, derived from external, isotopically depleted sources, metasomatized the granites, which resulted in Ce depletion as $Ce^{4+}$ and enrichment of the remaining REE, especially the HREE, contributing to formation of a globally important REE resource.

[1] Key Laboratory of Orogenic Belts and Crustal Evolution, School of Earth and Space Sciences, Peking University, Beijing 100871, China. [2] Department of Geology and Pedology, Mendel University, Brno 61300, Czech Republic. [3] Central European Institute of Technology, Brno University of Technology, Brno 61600, Czech Republic. [4] School of Environment and Technology, University of Brighton, Brighton BN41 2HQ, UK. [5] Institute of Geology and Geophysics, Chinese Academy of Sciences, Beijing 100029, China. [6] Institute of Mineral Resources, Chinese Academy of Geological Sciences, Beijing 100037, China. Correspondence and requests for materials should be addressed to C.X. (email: xucheng1999@pku.edu.cn).

The rare earth elements (REE) are critical materials because of their increasing importance in renewable energy and high technology applications. This, coupled with REE export decreases from China (which supplies more than 85% of the World's REE[1]), has led to a surge in REE exploration worldwide. Heavy REE (HREE: $Gd - Lu + Y$) are much rarer than light REE (LREE: $La - Eu$) because of their lower crustal abundances and limited reserves[2]. The most HREE-rich materials exploited in China are ion-adsorption clays, which supply the bulk of the World's HREE requirements[2]. The particular type of HREE ion-adsorption deposit studied here is thought to be formed by the leaching of HREE from granitic rocks, followed by fixing of the elements as ionic complexes adsorbed onto clays in the weathering crust. These deposits are currently only known in southern China, but comparable systems have been recently identified in Southeast Asia[3], Madagascar[4] and the Southeastern United States[5], where the deposits are generally enriched in LREE. From the perspective of HREE exploration, the vital question that needs to be addressed is why and how the HREE in the weathering crust are related to the parent granitoids, so that successful exploration criteria can be formulated. The current *situ*ation, where only one area appears to have produced large HREE deposits by this weathering mechanism, is enigmatic and requires further investigation.

It is generally accepted that REE were mobilized and fractionated during intense weathering of the granite under warm and humid conditions. The main mechanisms responsible for REE fractionation and enrichment during weathering are: preferential solubility of REE-containing primary minerals[6], complexation and the differential stability of the resulting aqueous complexes[7], sorption onto colloids and secondary minerals[8] and biological activity[9]. Most of the REE in granitoids tend to be concentrated in LREE-rich mineral phases such as apatite, monazite-(Ce) and fluorcarbonate minerals. Dissolution of these minerals would result in LREE enrichment. The formation of aqueous complex ions can also fractionate REE during weathering. HREE form stronger complexes with fluoride and carbonate relative to LREE at ambient temperature[7], thereby increasing the concentration of the HREE in solution and making them available for adsorption. However, studies show that groundwater and clay minerals are relatively enriched in HREE and LREE, respectively[10]. Secondary REE-rich minerals formed during weathering processes, some of which occur as colloids[11], would be adsorbed by clays in the weathering profiles. Studies of sample material have suggested that sorption on natural kaolinite preferentially retains LREE and results in strong LREE and HREE fractionation[8]. Experiments

have shown that kaolinite may preferentially adsorb the HREE in high ionic strength solutions (0.5 M), but such effects are negligible in low ionic strength solution (0.025 M)[12]. The upper clay-rich horizon from our drilling samples does not show high HREE concentration. Bacterial and fungal hyphae can fractionate REE through organic complexation and REE-rich mineral dissolution[9]. This process is generally limited to the upper weathering profile. Therefore, weathering may preferentially enrich the LREE and produce a high LREE/HREE ratio[8,10]. This process cannot really explain HREE-rich mineralization in South China. It is imperative, therefore, to determine and assess the origin of the HREE in the granites, and hence the role of chemical addition to the deposit formation.

Here we present data on previously unidentified, Ce-poor, REE minerals with both LREE and HREE enrichments in the primary granites from the Zhaibei REE-enriched clays, southern Jiangxi province. Most of the parental granitoids in the region formed in the Jurassic to Cretaceous periods[13]. The HREE-enriched clay deposits mainly occur south of latitude 28° N and more than 90% of the REE resources are located in Jiangxi, Guangdong and Guangxi provinces. The total reserves of this type of REE deposit are in the millions of tons of rare earth oxides and are particularly significant resources for HREE[14]. The weathering profiles, ranging from 5 to 30 m thick, are well preserved because the topography of the region is dominated by low hills with low denudation rates. Samples were collected from drill cores from the weathering profiles in Muzishan (MZS), Kaizidong (KZD) and Jiazibei (JZB) within the Zhaibei area (see Fig.1 in ref. 15).

## Results

**Chemical compositions**. The enrichment levels in LREE (La) and HREE (Y) from the weathering profiles do not increase with increasing depth (Supplementary Table 1; Supplementary Fig. 1). The highest La and Y abundances are found at the upper or lower part of this profile, respectively. The most weathered portions of the profiles show LREE enrichment with $La/Yb_{CN}$ (cn—chondrite-normalized) $> 3$, and an inconsistent and variable REE fractionation with increasing depth. Not all weathering profiles show HREE enrichment relative to LREE from surface downwards. Some have relatively constant $La/Yb_{CN}$ ratios with increasing depth. These weathering profiles contain a strongly negative to positive Ce anomaly with the highest positive values observed in upper and lower parts (Supplementary Fig. 1). The concentration of La and Y increases as Ce concentration decreases (Fig. 1).

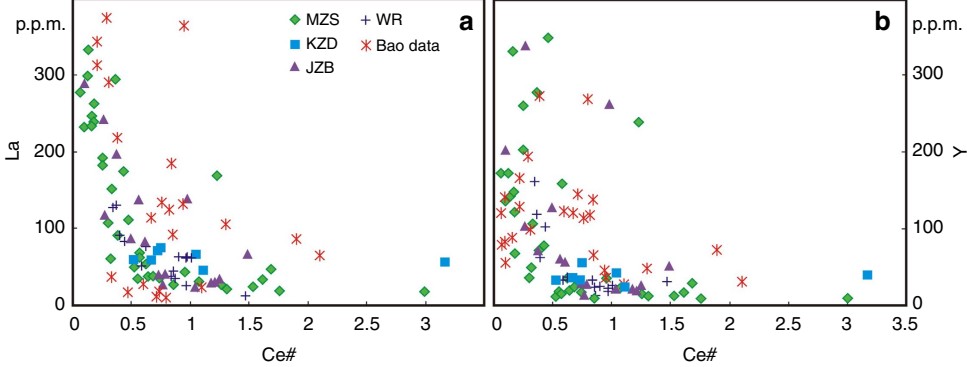

**Figure 1 | Diagram showing La and Y compositional change with Ce anomaly for weathering profiles and original whole rocks. (a,b)** Use La and Y as proxies for total light and heavy REE, respectively. Both La and Y concentration increases as Ce# (chondrite normalized Ce anomaly: $Ce_{CN}/(La_{CN} \cdot Pr_{CN})^{1/2}$) decreases. Coloured symbols: weathering profiles from MZS (Muzishan), KZD (Kaizidong), JZB (Jiazibei), and original whole rocks (WR) from corresponding sites, all in the Zhaibei area. Previous data are from weathering profiles in the Xinxiu, Guposhan, Huashan and Heling areas[14].

Zircons from the Zhaibei granites yield a concordant U-Pb age of 188 ± 0.6 Myr (Supplementary Table 2; Supplementary Fig. 2). The rocks mainly consist of K-feldspar, plagioclase, quartz and biotite (Supplementary Table 3). They have a chemical index of alteration[16] (CIA: molar $Al_2O_3/(Al_2O_3 + Na_2O + K_2O + CaO) \times 100$) $\sim54$ (Supplementary Table 4), similar to typical fresh granitoids with CIA of 45–55 (ref. 16) and lower than the weathering profiles in South China (CIA $>70$)[14]. Importantly, higher REE enrichment of the original granites is also strongly related to stronger depletion in Ce (Fig. 1). Therefore, the Ce anomaly may be a good indicator of degree of the HREE enrichment.

**REE minerals in granites.** REE minerals are relatively abundant in the granites as phosphates and fluorcarbonates (Fig. 2a, Supplementary Fig. 3 and Supplementary Table 3). Monazite-(Ce) mostly occurs as inclusions in biotite and ilmenite. Minor monazite-(Ce) is associated with Ca-REE fluorcarbonates, zircon and rutile, being a pseudomorph of an earlier REE-Zr-Ti mineral (mosandrite?). Xenotime-(Y), while present, is not common, and occurs as radiating spines and densely packed aggregates around zircon and thorite. Primary fluorapatite (up to 1 mm in length) is

the dominant REE-bearing mineral present. It is slightly enriched in LREE with La/Yb$_{CN}$ $\sim2$ and a negative Eu anomaly (Supplementary Table 4). The Ca-REE fluorcarbonates are Ce dominant and contain high $Y_2O_3$ (up to 5.3 wt%). Minor calcite is observed to intergrow with feldspar and quartz in the granites (Fig. 2b). It contains high SrO abundance ($\sim1.8$ wt%; Supplementary Table 3), which is a rare occurrence for granite-hosted calcite.

Three unidentified HREE-rich minerals (REE-1, 2, 3) are found in the granites (Fig. 2c–f; Supplementary Table 3), where the fluorapatites have been altered to form new REE minerals. Under back scattered electron (BSE) imaging, the altered parts show irregular patches with variable REE concentrations, and with two dominant REE compositions (REE-1 and REE-2), that is, Nd and Y dominant, respectively. With decreasing brightness under BSE imaging (Fig. 2c,e), the $Y_2O_3$ abundance increases from about 10 to 51 wt%, and $La_2O_3$ and $Nd_2O_3$ decrease from 12 and 20 wt% to $<1$ wt%, respectively. The REE-1 phase shows relatively low F ($\sim1$ wt%) and variable $La_2O_3$ (4.5–12 wt%) and $Y_2O_3$ (10–36 wt%) compositions. The REE-2 phase (Y-dominant phosphate) has a higher F content (6–9 wt%). Both REE-1 and REE-2 have consistently low Ce contents, close to or below the detection limit (Figs 3 and 4a). The Ca-poor REE fluorcarbonates (REE-3) are also heavily depleted in Ce and occur as

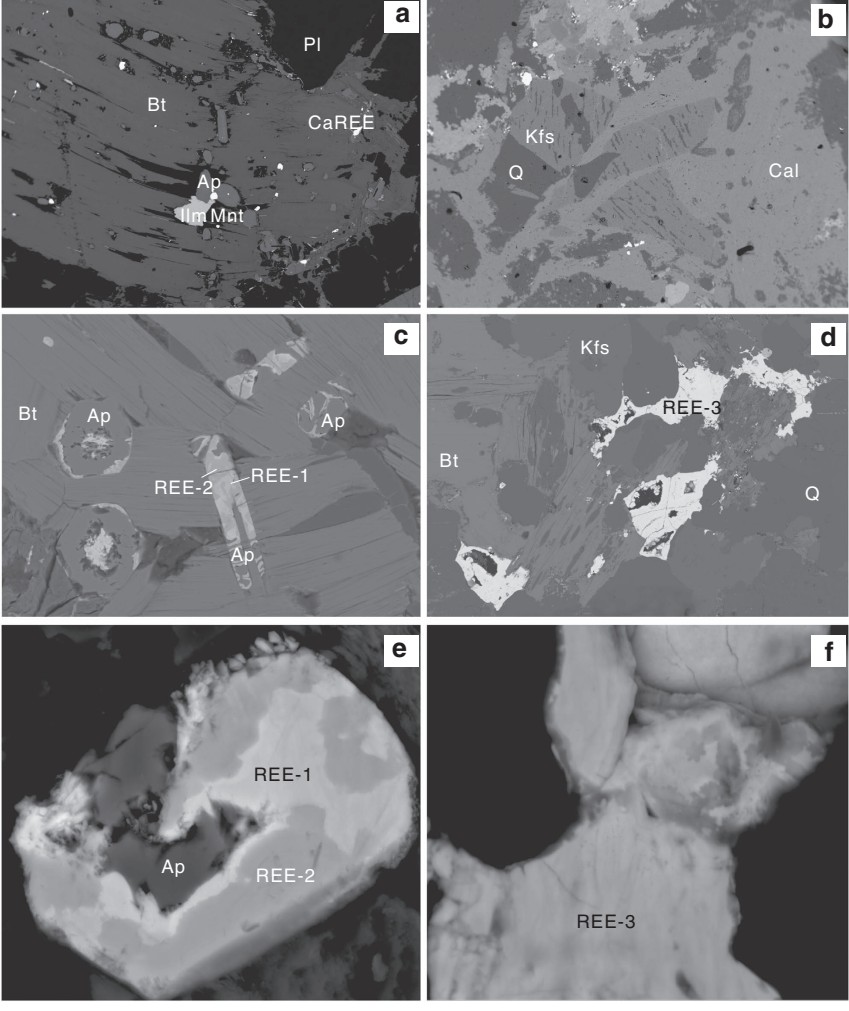

**Figure 2 | Selected characteristic textures observed in the granites. (a)** Primary REE minerals including Ca-REE fluorcarbonate (Ca-REE) and monazite-(Ce) (Mnt) associated with fluorapatite (Ap) and ilmenite (Ilm) in biotite (Bt). **(b)** Sr-rich calcite (Cal) inter-grown with K-feldspar (Kfs) and quartz (Q). **(c)** fluorapatite altered to produce REE phosphate minerals REE-1 and REE-2, which contain high HREE and Ce-poor compositions. **(d)** anhedral interstitial REE fluorcarbonate (REE-3) with high HREE and low Ce abundances. **(e,f)** high magnification images of the unidentified REE minerals. Pl, plagioclase. Field of view: **(a)** 3 mm; **(b)** 2.5 mm; **(c)** 0.15 mm; **(d)** 1.5 mm; **(e,f)** 30 µm.

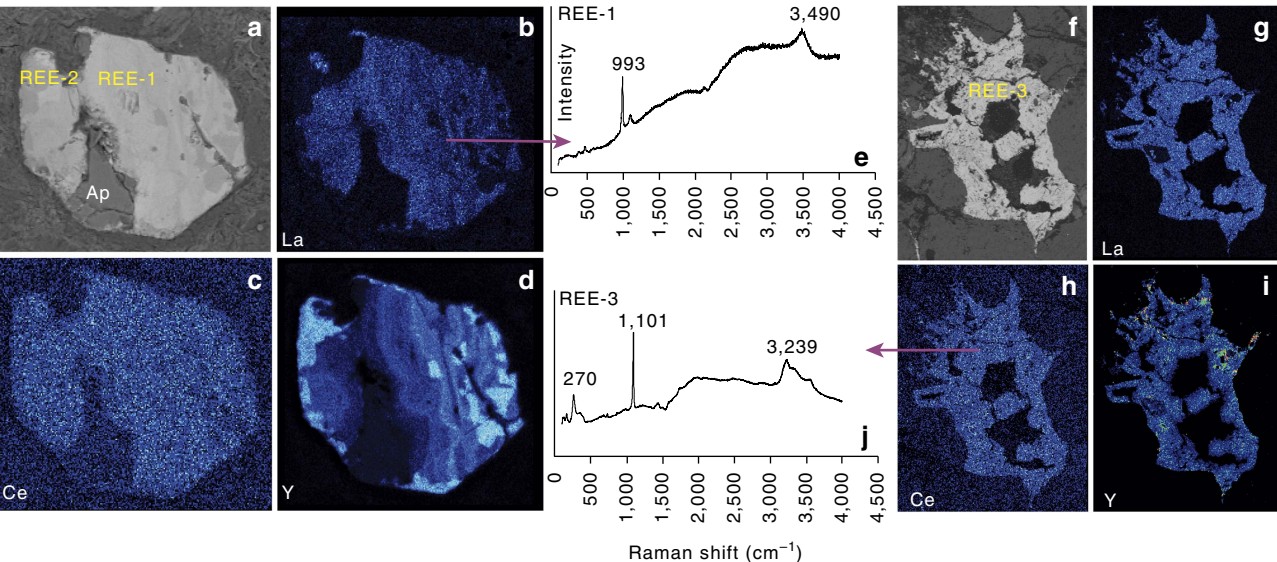

**Figure 3 | REE compositional maps and Raman spectroscopic analyses of altered fluorapatite and REE fluorcarbonates.** (**a–e**) REE phosphates (REE-1 and REE-2); (**f–j**) REE fluorcarbonates (REE-3). (**a,f**) backscattered-electron images (field of view: 60 and 350 μm, respectively). (**b–d,g–i**) X-ray distribution maps of La, Ce and Y. (**e,j**) Raman spectroscopic data showing the occurrence of water in the mineral structure (3,490 cm$^{-1}$ and 3,239 cm$^{-1}$ for REE-1, REE-3, respectively).

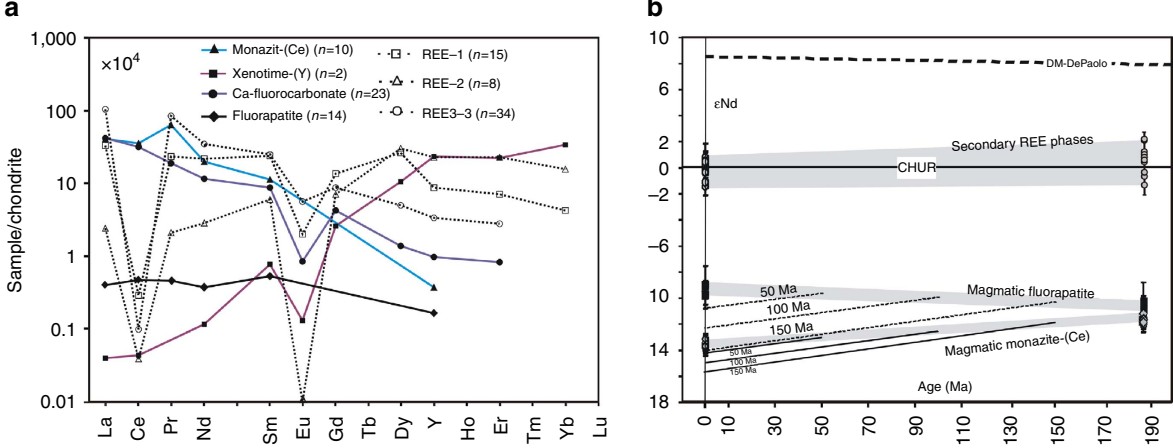

**Figure 4 | Chondrite-normalized REE patterns of REE minerals and $\varepsilon_{Nd}$ values for magmatic REE-phosphates and secondary REE minerals at the time of granite crystallization and present day.** (**a**) Average REE content of REE minerals. (**b**) Neodymium isotope data indicate different REE sources in primary and secondary minerals, rather than derivation of the REE in secondary minerals via a two-stage Nd isotope evolution model[50]. The two-stage model is shown by the fine lines, and assumed the alteration of monazite-(Ce) and fluorapatite, releasing Nd to give the secondary phases. The curves show the evolution of $\varepsilon_{Nd}$ through radiometric decay for weathering events which fractionated Sm and Nd to the elemental ratios in the secondary minerals at 150, 100 and 50 Myr ago, respectively. The calculations used the decay constant and equations given by ref. 51. Release of REE from monazite-(Ce) or fluorapatite breakdown during weathering cannot explain the $\varepsilon_{Nd}$ in secondary phases, implying that the secondary phases derived their REE from a separate, isotopically depleted, source. CHUR, chondritic uniform reservoir.

disseminated, anhedral grains interstitial to biotite, feldspar, and quartz. *In situ* laser-ablation ICP-MS analysis shows that the Ce content is <500 p.p.m. (Supplementary Table 4). The REE-3 phase is La, Nd and Y dominant, and has a higher total REE, especially La$_2$O$_3$ (up to 36 wt%) and Y$_2$O$_3$ (up to 10 wt%) than normal bastnäsite-(Ce), which is the dominant LREE mineral in nature. The Raman spectrum indicates the presence of H$_2$O in the REE-1 and REE-3 (Fig. 3e,j).

These HREE-rich minerals are more abundant than xenotime-(Y) in the granites, and are a significant source for the HREE mineralization in the late weathering process. They represent the first report of REE phosphates and carbonates with strong depletion in Ce relative to La and Pr (Fig. 4a). This suggests that

these minerals formed under high oxidation conditions. The oxidation of the Ce ion from 3$^+$ to 4$^+$, compared with the other REE ions which can only have a 3$^+$ charge, is responsible for the anomalous behaviour of Ce. Experiments show that an oxygen fugacity ($f_{O2}$) greater than the fayalite-magnetite-quartz (FMQ) buffer is required for a Ce$^{4+}$/$\sum$Ce ratio <1 (ref. 17). The log(Ce$^{4+}$/Ce$^{3+}$), and consequently log$f_{O2}$, have a linear dependence on $1/T$ (ref. 18).

**Source of REE.** The Sr isotope of Sr-rich calcite and Nd isotope of primary monazite-(Ce) and fluorapatite and secondary REE-1 and REE-3 minerals in the granites were analysed *in situ* (Fig. 4b;

Supplementary Table 5). Given the extremely low $^{87}Rb/^{86}Sr$ ratio ($< 0.002$) of the calcites, their measured $^{87}Sr/^{86}Sr$ ratio is considered to approximate the initial ratio. Calcite has lower initial Sr isotope ratios (0.7054–0.7061) than fresh feldspar (0.7088–0.7241) in the rocks analysed by solution (Supplementary Table 5), and reported Zhaibei granites ($\sim 0.7110$)[19]. The Ce-poor, HREE-rich minerals have a higher $\varepsilon Nd(t)$ ($0.9 \pm 0.8$) than the primary LREE-rich minerals, which have a consistent negative $\varepsilon Nd(t)$ ($-11.5 \pm 0.5$). In contrast, the feldspar contains lower Nd isotope ratios ($\varepsilon Nd(t) = -2.2$ to $-13$). The granites in the Zhaibei area also show a wide range in $\varepsilon Nd(t)$ ($-0.8$ to $-7$)[19], again lower than the HREE-rich minerals. It is difficult to accurately obtain the initial Nd isotope composition of these secondary REE minerals because of unknown mineral age and variable Sm/Nd ratio in an open environment. On the basis of mineral chemical data and calculated isotopic evolution trends for different weathering periods, the metasomatic fluids inferred to be responsible for the HREE enrichment show a more depleted source than the granites, from the time of granite crystallization (188 Myr ago) to present day (Fig. 4b). Regardless of the age of secondary mineralization, the REE in secondary phases cannot be sourced from the host rocks.

## Discussion

The most important factor in the Zhaibei weathering profiles in terms of scientific and economic interest is their enrichment in the HREE. Heavy REE enrichment is commonly observed in peralkaline and alkalic granitoids, where it is related to the metasomatism of zirconosilicates[20,21]. The alkali-rich composition is believed to increase the solubility of zircon and volatiles in the melt phase[22], which in turn enhances the mobility of REE. Therefore, alteration of HREE-containing zircon by hydrothermal fluids leads to the formation of minor xenotime-(Y) and HREE-silicates as small anhedral grains and micro-inclusions around or in the zircon[20,21]. Similar textures, involving xenotime-(Y) associated with zircon, have been observed in the Xihuashan granites from South China[23].

However, our mineralogical and geochemical data from the Zhaibei granites indicate alteration and mineralization by an oxidized, REE-rich metasomatic fluid with low Sr and high Nd isotope ratios, which does not support derivation of the REE from the primary granite. Experiments indicate that the REE do not preferentially partition into fluids exsolved from granite magmas[24]. The titanite + magnetite + quartz assemblage in the granites (Supplementary Fig. 3d) indicates oxygen fugacities of $\sim 1$ ($\Delta$FMQ) at a magma temperature of 800 °C (refs 15,25), which is too low to oxidize $Ce^{3+}$ to $Ce^{4+}$; (refs 17,18). Circulating meteoric water could contain high oxygen, and possibly REE derived from dissolution of early crystallizing REE minerals in the granites during weathering, but it would inherit the high Sr and low Nd isotope compositions of the host rocks. Fluids derived from the surrounding basement rocks would also have strongly high Sr isotope ratio and negative $\varepsilon Nd$ value[26], and so are unlikely to be responsible for the secondary REE mineralization.

The Zhaibei rocks show similar chemical compositions to A-type granites[15]. This implies a mantle contribution to the granite origin. However, the oxygen fugacity of the upper mantle falls in $\pm 2$ log units of the FMQ oxygen buffer[27]. The fluids in mantle peridotite xenoliths have the $f_{O2}$ of near the FMQ at 1,200 °C and 10 kbar[28], hence mantle-derived fluids are unlikely to account for the Ce-depleted mineralization. Subducted slab-derived melts/fluids are a candidate because of their high oxygen fugacity and $\varepsilon Nd$ value. Geophysical data and geological and geochemical characteristics suggested Paleo-Pacific plate subduction underneath Mesozoic South China[13,29]. It is well

known that Si-rich melts derived from the subducted slab can be tonalitic-trondhjemitic-granodioritic or adakitic[30,31]. The silica-saturated melt will react with mantle peridotite to induce mantle and resultant crustal melting. Adakitic melt is considered as a candidate for the source magmas for Mesozoic porphyry Cu deposits in South China[32], because it has high oxygen fugacity which can eliminate sulfides in the mantle source[33]. However, adakitic melts may be not a key metasomatic agent for the HREE mineralization because they are strongly depleted in HREE[34], and there are presently no known REE deposits associated with adakitic rocks.

In contrast to silicate melts, aqueous fluids can transfer REE in subduction zones. Experiments show ligands such as $F^-$, $Cl^-$, $CO_3^{2-}$ significantly enhance REE solubility in slab-derived aqueous fluids, and addition of $CO_3^{2-}$ results in preferential solubility of the HREE[35]. A flat-slab subduction model was proposed for the formation of a broad intracontinental orogen and large scale magmatic province in Mesozoic South China[13]. Mungall[33] suggested two types of fluids formed during flat subduction. If a slab passes under the overriding plate for a long distance at a low angle, then falls back down into hot asthenospheric mantle, the result would be the release of aqueous fluids rather than adakitic magmas. If there is no asthenosphere present above the slab, then it would follow a cold path and only produce aqueous fluid without fluxing of the mantle. Moreover, the seafloor sediments contain significantly high REE abundance (up to 2,200 and 1,100 p.p.m. in the eastern Pacific Ocean and eastern Indian Ocean, respectively)[36,37]. Melting of a sediment layer on the subducting slab has a high potential for REE mineralization[38,39]. Experimental investigation shows that melting of carbonate-bearing oceanic crust can form a carbonate melt/fluid as primary carbonatite magma[40], which is distinct from mantle-derived carbonate melts with low REE abundance[41] that must undergo strong fractionation in the crust to produce REE enrichment in surface carbonatites[42]. Carbonate melts/fluids can contain 10 wt% water[43] indicating a high oxygen fugacity, and have extremely low viscosities[44] facilitating their rapid ascent through deep faults to react with the granites. Importantly, Sr-rich calcite was found in the Zhaibei granites (Fig. 2b), and shows relatively low Sr isotope ratios, similar to calcite of a typical carbonatitic origin[45,46].

On this basis we propose the following model for HREE mineralization in South China. Early Jurassic A-type granite magmas occurred in South China, indicating the ending of orogeny and beginning of slab break-off[13]. Heavy REE-enriched secondary minerals and calcite occur in otherwise unaltered granites at the base of weathering profiles, but the isotopic compositions of these minerals are not consistent with the derivation of REE and Sr from the host granites, or currently exposed country rocks. We cannot rule out leaching of HREE from weathering of an unexposed isotopically depleted source rock, but such a source has not been identified. Alternatively, during the formation of the plutons hydrous, carbonate-rich fluids released from the subducted slab underneath Mesozoic South China, characterized by high oxidation state and REE contents, metasomatized the granites. This resulted in the oxidation of $Ce^{3+}$ into $Ce^{4+}$ while not affecting the other REE, resulting in the formation of HREE-rich phosphates and carbonates with negative Ce anomalies. The weathering profiles inherited the REE signature of the granites, reflecting the depletion in Ce and enrichment in both LREE and HREE (Fig. 1). Critically, for current global resource requirements, either model implies that tropical weathering may not universally result in HREE-enriched secondary mineral deposits. Exploration models must incorporate identification of HREE enrichment mechanisms within the protoliths of deposits formed by

weathering process to successfully target HREE deposits. The identification of large scale tectonic controls on the geochemistry of primary REE-rich fluids is the key in this process.

## Methods

**Major and trace element analysis.** Major elements of the granites were determined using wet chemical methods at the Institute of Geochemistry, Chinese Academy of Sciences (CAS). Trace elements of the rocks and weathering crusts were analysed in solution by ICP-MS at the Institute of Geochemistry, CAS. Details are given in ref. 47. The analytical precision for most elements is generally better than 10%. *In situ* laser-ablation (LA) ICPMS analyses of primary fluorapatite and secondary REE-1 and REE-3 from the granites in polished thin section were performed at the School of Earth and Space Sciences, Peking University. The diameter of the ablation spot is 32 and 24 μm for apatite and REE minerals, respectively. NIST 610 glass was used as a calibration standard for all samples. The element used for the internal standard was Ca and La, expressed as CaO and $La_2O_3$ for apatite and REE minerals, respectively. Analytical error is $\leq 5\%$ at the p.p.m. level. In-run signal intensity for indicative trace elements was monitored during analysis to make sure that the laser beam stayed within the phase selected and did not penetrate inclusions.

**Zircon U-Pb dating.** Zircon grains for U-Pb dating were picked using conventional magnetic and density techniques, and then individual grains were hand-picked. Transmitted and reflected light photomicrographs and cathodoluminescence images were used to observe internal structures of the zircons. Age determination was done using LA-ICPMS at the School of Earth and Space Sciences, Peking University. For LA-ICPMS measurements, we used an Agilent 7500ce mass-spectrometer coupled to a 193-nm ArF excimer laser. The laser beam was focused on the sample with a fluency of 20 J cm$^{-2}$ and a spot of 32 μm in diameter at a repetition rate of 5 Hz for 40 s. Helium was used as a carrier gas to transport the ablated aerosol to the mass-spectrometer. Zircon Nancy 91500 was used as an external calibration standard to correct for instrumental mass bias and elemental fractionation. Standard PLE were employed for quality control. The Pb content of zircon was externally calibrated against NIST 610 with Si as an internal standard, whereas other trace elements were measured with Zr as an internal standard. Raw-data reduction was performed off-line using the Glitter software.

**Mineral analysis.** The major-element compositions of mineral phases were determined by wavelength-dispersive X-ray spectrometry using a Cameca SX100 electron microprobe at the Masaryk University, Czech Republic, and an EPMA-JXA-8230 electron microprobe at the Key Laboratory of Metallogeny and Mineral Assessment, Chinese Academy of Geological Sciences. For each mineral or group of minerals, their own set of appropriate matrix-specific standards (both natural and synthetic) and optimal instrumental conditions (beam settings, detector type and counting statistics) were carefully chosen by performing multiple measurements. The microprobe was operated at an accelerating voltage of 15 kV and a beam current of 20 nA, with an electron beam defocused to a 5–10 μm spot to limit devolatilization, ionic diffusion and other forms of beam damage in the samples. For REE-bearing minerals, raw wavelength-dispersive X-ray spectroscopy (WDS) data were corrected using empirical interference values for REE and other elements potentially interfering with the REE signals determined from the well-characterized synthetic standards (glasses and orthophosphates). These corrections decreased the limit of detection to the 1,000 p.p.m. level (Pr, Nd, Gd, Dy and Yb) or better ($<600$ p.p.m. for La, Ce and Sm). Compositional X-ray maps of REE minerals were obtained with an Oxford INCA X-MAX50 250 +, energy-dispersive X-ray spectrometer installed on a FEI Quanta-650FEG scanning electron microscope, at the School of Earth and Space Sciences, Peking University. The back-scattered electron and energy-dispersive X-ray data acquired from the sample were combined and processed automatically to generate the most sensitive X-ray mapping. The sample, coated with a conductive Cr layer (10 nm thickness) to prevent sample charging, was analysed in a high-vacuum mode at standard operating conditions (accelerating voltage of 20 kV, probe current 5 nA).

**Raman analysis.** Raman spectroscopy of REE minerals was performed on a Renishaw-inVia Reflex Laser Raman microprobe using the 532 nm line of laser at the School of Earth and Space Sciences, Peking University. The laser spot size was focused to 1 μm. Accumulation times varied between 5 and 10 s. The estimated spectral resolution was 1 cm$^{-1}$ and calibration was performed using synthetic silicon.

**Isotopic analysis.** The Sr and Nd isotopes of the fresh feldspars were analysed by a MC-ICPMS (VG AXIOM) at the School of Earth and Space Sciences, Peking University. Mass fractionation corrections for the Sr and Nd isotope ratios were normalized to $^{86}Sr/^{88}Sr = 0.1194$ and $^{146}Nd/^{144}Nd = 0.7219$, respectively. Repeated measurements for the Nd standard JNdi and Sr standard NBS987 yielded $^{143}Nd/^{144}Nd = 0.512120 \pm 12$ ($2\sigma$) and $^{87}Sr/^{86}Sr = 0.710252 \pm 12$ ($2\sigma$), respectively. For calculating initial Sr–Nd isotope values, the feldspar Rb, Sr, Sm, Nd

concentrations were analysed by solution ICPMS (Thermo Fisher Scientific X-Series II). The data processing procedure included linear drift correction, internal (matrix) correction, REE and Ba interference corrections, blank subtraction, calibration with international standards and a dilution correction. *In situ* Sr isotope of calcite and Nd isotope of REE-rich minerals in polished thin section were determined by a Neptune LA-MC-ICPMS at the Institute of Geology and Geophysics, CAS. The laser was focused at 40–60 μm for calcite, 120 μm for primary fluorapatite and 24 μm for REE-rich minerals, and fired using a 6–8 Hz repetition rate and an energy density of 10 J cm$^{-2}$. Standard apatite AP1 and coral were determined as external calibration for calcite, and apatites AP1, AP2, Outter Lake and Jefferson monazite 44069 reference materials for REE-rich minerals. Before laser runs, the MC-ICPMS was tuned using a standard solution to obtain maximum sensitivity. Normalized Sr and Nd isotope ratios were calculated using the exponential law[48,49].

**Data availability.** The authors declare that all relevant data are available within the article and its Supplementary Information files.

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

## Acknowledgements

This research was financially supported by Chinese National Science Foundation (Nos. 41573033, 41222022), European Union's Horizon 2020 (No. 689909) and Czechic Project CEITEC 2020 (LQ1601). We thank Liang Qi for help with trace element analysis, and Daniel Harlov for reading and commenting on an early version. We are particularly grateful to Sebastian Tappe and two anonymous reviewers for reviewing and improving the manuscript.

## Author contributions

C.X., J.K., Z.Z., C.H. and W.S. were responsible for sample collections. C.X., J.K., A.K., Y.Y., W.S., Z.Z. and C.H. were responsible for element and isotope analyses. C.X., J.K., M.B. and T.U. were responsible for mineralogy and petrology of granites. C.X. and M.P.S. worked on data interpretation. C.X. wrote the manuscript with assistance from M.P.S.

## Additional information

**Competing financial interests:** The authors declare no competing financial interests.

