## [Peer Review File · Nature Communications]

Reviewers' Comments:

Reviewer #1 (Remarks to the Author)

Comments for Author

This paper describes an interesting and potentially significant advance in understanding the origin of a type of heavy rare earth element deposit developed in granite-derived regolith. This paper will be of particular interest to researchers working on both granite petrogenesis and granite-related REE resources. The authors identify a number of previously unrecognized HREE-enriched and Ce-depleted minerals (2 phosphates and 1 fluorocarbonate) in the Zhaibei granite. The new mineral chemistry and Nd isotope data are used to infer unique aspects of the petrogenesis of the granite from which the globally important Ion Adsorption Clay (IAC) deposits at Zhaibei formed.

The main thesis of the paper is focused on a comparison of the mineral chemistry of the primary and secondary mineral phases. The authors make key assumptions in this work: (1) measured Sm/Nd ratios of the secondary phases are reasonably accurate, 2) all relevant primary hosts of REE (esp. Nd, Sm) have been identified, and 3) thus, there are no other primary magmatic REE-bearing minerals that could have contributed to the Sm/Nd and isotopic signatures. Assuming the above, this paper describes a potentially significant process that may be critical to the generation of HREE IAC deposits.

The paper also highlights a new area of fundamental research (combining textural, mineral, and isotopic data) needed on this important deposit type. I recommend publication of this paper.

In order to more accurately reflect research on this topic, I also recommend that the authors should include reference to the following recently published studies: Foley et al., (2015b, 2015b), Ishihara et al (2008), Janots et al. (2015). In all of these cases, the studies also focused on very similar geochemical aspects and origins of this deposit type (REE petrogenesis of granites hosting REE IAC deposits, Ce evolution in REE-enriched laterites, and REE complexation under surficial conditions).

Here are a few minor comments to clarify the text:

Specific Comments by line:

Some of the English is non-standard, with sentences missing articles and having awkward phrasing. This can be easily corrected.

Line 0: Detailed datasets, complete descriptions of the methodology, and additional figures are included in the supplementary information section. Tables S2 (compositions of mineral in the granite) and S3 (major and trace compositions of granite and fluoroapatite and secondary REE minerals) mix whole rock data and mineral chemistry. Symbols in Figure 4a are not decipherable (too small, and lack contrast). Are color figures possible?

Line 40-42: The authors should cite additional recent papers describing REE IAC occurrences in Madagascar and the Southeastern United States (as suggested here).

Example 1, Janots et al., 2015, discuss a REE-enriched lateritic profile wherein Ce is not necessarily oxidized nor fractionated from other REE during weathering in lateritic conditions, suggesting that like Ce(III), Ce(IV) can be mobilized in aqueous fluids during weathering, possibly due to complexation with organic molecules.

Janots, E., Bernier, F., Brunet, F., Munoz, M et al. (2015) Ce(III) and Ce(IV) (re)distribution and fractionation in a laterite profile from Madagascar: Insights from in situ XANES spectroscopy at the Ce LIII-edge: *Geochimica et Cosmochimica Acta* 153, p. 134-148/

Example 2, Foley and Ayuso, 2015, describe potentially comparable L-M REE enriched regolith deposits with up to 1500 ppm readily extractable REE in weathered granites of the southeastern United States. This may be the first example of an occurrence of REE-IAC in granite-derived regoliths to be identified outside of China and Southeast Asia.

Foley N., Ayuso R., Hubbard B., Bern C., Shah A. (2015) Geochemical and Mineralogical Characteristics of REE in Granite-Derived Regolith of the Southeastern United States: Proceeding of the 13th Biennial Society for Geology Applied to Ore Deposits (SGA) Meeting, Proceedings, Volume 2, p. 725-728. <https://www.researchgate.net/publication/281406501>)

Line 68-69: The Author should cite a reference(s) for the statement describing bacterial fractionation of REE through organic complexation.

Line 100-102: The Authors may want to include a plot to illustrate the described relationship between bulk REE (concentrations) in the granite and Ce anomaly.

Line 101: A recent paper by Foley and Ayuso (2015) describe some REE characteristics of granite hosts that may also be evident in REE IAC deposits of the Southeastern United States. They note the presence of REE tetrad behavior in many examples of granites that host IAC deposits. The data sets included in that paper show the presence of a negative Ce anomaly in granites of Xinzui HREE IAC and Zudong HREE IAC (including data from Bao and Zhou, 2008). It would be useful to add a plot showing the REE pattern for the ZhaiBei granites and describe whether similar tetrad behavior is present as this may support extreme fractionation and/or late magmatic - hydrothermal alteration, processes that are discussed as important in the petrogenesis of REE-enriched granite source rocks.

Foley, N. and Ayuso, R. (2015) REE enrichment in granite-derived regolith deposits of the Southeastern United States: Prospective source rocks and accumulation processes. In: Simandl, G.J. and Neetz, M., (Eds.), British Columbia Ministry of Energy and Mines, British Columbia Geological Survey Paper 2015-3, pp. 131-138

<http://www.empr.gov.bc.ca/Mining/Geoscience/PublicationsCatalogue/Papers/Documents/P2015-3/16%20Foley.pdf>

Line 109: These are remarkably large 'igneous' apatite crystals. Wang et al., 2015, describe the mineral suite for ZhaiBei granite as including allanite, as well as monazite and apatite. It is noteworthy that igneous apatite is suggested to be the primary carrier of REE, rather than allanite. Late magmatic-hydrothermal alteration of igneous allanite can lead to formation of monazite, zoned apatite crystals + bastnäsite + calcite. Is there any evidence for REE-enriched overgrowths on apatite as has been described by Ishihara et al, 2008?

Ishihara, S., Hua, R., Hoshino, M., Murakami, H., 2008. REE abundance and REE minerals in granitic rocks in the Nanling range, Jiangxi Province, Southern China, and generation of the REE-rich weathered crust deposits. *Resource Geology* 58, 355-372.

Line 118: Change to 'metasomatized to form new REE minerals', or 'new phosphate and fluorocarbonate' (see Line 128).

Line 122-125: The paper could be improved by the addition of higher magnification photographs of the unknown mineral phases so that the described features could be better seen. This is an important addition because many of the arguments depend on the identification of the 'new' secondary REE minerals. Although the mineral data is likely insufficient at the present time to meet IMA standards for new mineral characterization, the paper raises an awareness of the need for further studies to identify and characterize unknown REE mineral phases in these systems.

Line 139-143: Provide appropriate reference(s) to the experimental study described.

Line 141-145: Data for fluoroapatite appears to be missing from Figure 4a. Symbols are difficult to distinguish.

Line 146-149, and in Discussion:

Comment on the impact of the breakdown of allanite (Wang et al, 2015) as a contributing mineral to the Nd budget in the bulk granite. Consider including Nd isotope data for the allanite.

Contrast the Nd isotope results of this study to the Nd isotope study described by Foley and Ayuso (2015) for a less deuterically altered A-type granite where a 'moderate scatter in Nd isotope ratios and Sm/Nd values in the surficial materials, compared to bedrock' was found to be simply consistent with mineral weathering, whereas Nd isotopes and Sm/Nd values of the extractable ions suggested weathering of a mineral having bulk compositions comparable to bedrock (in that case allanite was in the granite was not altered to intermediate phases such as apatite, bastnäsite, or monazite).

I strongly recommend publication of this paper, particularly if the main thesis of the work is comparing the mineral chemistry of the in-situ primary and secondary mineral phases, and assuming the authors have 1) reasonably accurate Sm/Nd ratios, 2) identified all important host of REE (esp. Nd, Sm) of the secondary phases, and 3) there are no other primary magmatic REE-bearing minerals that could have yielded Sm/Nd signatures in the calculated ranges. In summary, the authors have explained the potential significance of a process that may be critical to generating HREE IAC deposits in China (with wide global implications for the origin of these deposits) and highlighted an important focus of further research on this important deposit type.

Reviewer #2 (Remarks to the Author)

This manuscript reports new HREE-enriched mineral phases from granites in southern China. The authors characterize these new phases by means of classic petrography, EPMA, Raman, and in situ Nd isotope analysis. They conclude that these phases formed by metasomatic reactions between the cooling granite body (REE-phosphates etc.) and invading HREE-enriched CO₂-bearing fluids/melts derived from the degassing Paleo-Pacific oceanic slab during the Mesozoic. These secondary metasomatic granitic HREE phases contributed to the weathering profiles that developed during tropical weathering and created world's largest HREE deposits in southern China.

These findings are interesting and certainly bear some novelty in explaining HREE enrichment in weathering profiles, a process that has remained enigmatic to date. The subject is topical and economic geology is very important to society (although underappreciated), so it certainly deserves a place in a Nature Publishing Group journal.

I am surprised that no XRD data are reported, but the newly reported phases may be too rare to be detectable in rock powders (how abundant are they?). As mentioned by the authors, the initial Nd isotope ratios are somewhat difficult to determine given the changing parent-daughter element ratios during the envisaged metasomatism. The undertaken Nd isotope modeling appears sound. I would have preferred to see another isotope system showing similar results such as Rb-Sr, or Lu-Hf (note that Lu is a HREE and this could have produced interesting results).

The literature references on carbonate-rich magmas or carbonatites could be updated; the work by Woolley is a bit outdated. There is an increasing body of evidence that carbonate melts in Earth's mantle are actually not that enriched in incompatible elements including the REE (see Foley et al., 2009, Lithos). This enrichment must also happen via zone refining and fractionation likely in the crust. For exceptionally REE enriched carbonatites and carbonate-bearing ultramafic silicate rocks (all mantle-derived) see tables in Tappe et al. (2006, Journal of Petrology).

The manuscript is generally well written, but it still contains a few minor grammatical errors and awkward phrasing that should be taken care of during the revisions. Figure 1 is missing explanations of the legend, even in the caption. The panels in Figure 3 are arranged in a very awkward way. Figure 4a is a bit messy; Figure 4b is simple but powerful.

I wish the authors good luck with the revisions,

Sebastian Tappe
University of Johannesburg

Some additional comments:

Line 32-35: I fail to see the logic of that statement; some rephrasing may help.

Line 129: it must say "laser-ablation MC-ICP-MS"; laser alone is not doing anything.

Line 137: check grammar.

Line 139, 177, 187: when reference is made to oxygen fugacity, it should be stated what the reference buffer is, i.e. FMQ or NNO or HM?

Line 149-152: why not collecting extra Rb-Sr or Lu-Hf isotope data? This would be comforting for the rather big conclusions drawn.

Line 157: check grammar.

Line 188: Not all subducted material is high in epsilon Nd; i.e. think of marine sediments in proximity to cratons. Maybe rephrase.

Line 191-192: unclear; what has the subducted slab to do with the subcontinental mantle? Or do you mean asthenosphere or convecting mantle?

Line 198: check grammar.

Line 217: Carbonatite references could be updated.

Line 226: Awkward. A magma cannot be metasomatized, but a rock or crystal mush can.

Line 234: It is a deposit due to weathering processes, but not a "weathered deposit", correct?

Reviewer #3 (Remarks to the Author)

General Comments

The origin of the mineralisation in the South China ion adsorption HREE deposits is a subject of considerable interest, considering that these deposits are currently the source of much of the World's supply of HREE. These deposits have received considerable media attention partly because of their economic importance and also because of the negative environmental impact of their exploitation. In this manuscript, the authors seek to explain the source of the HREE. They make a convincing case that the HREE originated from unidentified HREE-rich secondary metasomatic minerals in the adjacent granites. To my knowledge, although it is widely believed that the granite are the source of the HREE, the source within the granites has not previously been identified. The authors then go on to discuss the origin of the fluids, which they argue, on the basis of Nd isotopic data, introduced the REE. They make a reasonable case that these fluids originated during subduction slab melting. This interpretation also makes sense. Unfortunately, the manuscript makes large numbers of unsubstantiated statements or statements that are simply incorrect, as will be evident from the comments keyed to the text, and calls on sources that are inappropriate for making their case. In general, it needs to be written a lot more tightly. The quality of the writing is also well below that expected in an international journal. The manuscript will therefore require major revision before it is suitable for publication.

Comments keyed to the text

T

he term "superlarge" is inappropriate in the title.

Line 19. What about the other LREE, e.g., La and Nd. The statement here might apply to Ce because it is fairly immobile in the 4+ state but would not apply to the other LREE.

Line 23-25. The conclusion of this sentence does not follow from the preceding sentence. What is the basis for proposing that "REE-rich fluids.....metasomatized the granites during solidification"?

Line 51. For greater accuracy replace "solubility" with "REE" and "mobility" with "stability".

Line 55. What about fluorocarbonate minerals? They are referred to on line 104.

Line 53. Delete "role of".

Line 69. Supply a reference to support this statement.

Line 74. Why would it not be sufficient to determine the compositions of the granites? Why is it necessary to know the composition of the corresponding magmas?

Line 75. Why is Ce emphasized? Presumably concentrations of the other LREE are also low.

Line 92. The statement "The chondrite normalized (CN) REE patterns....." needs to be linked to a figure illustrating the profiles.

Line 98. Ditto.

Line 100. This statement, which is linked to Figure 1, can only apply to La and Y not REE generally.

Line 104. The IMA accepted name for sphene is titanite. REE minerals need to be given a suffix indicating the dominant REE. Thus, for example, monazite in Table S2 should be monazite-(Ce) Correct Table S2 accordingly.

Line 105. See preceding comment about REE mineral names. Correct here and elsewhere.

Line 135. How abundant are these unidentified REE minerals relative to xenotime-(Y), which is another potential source of HREE?

Line 138. Explain the statement "This suggests that these minerals formed under high oxidation conditions".

Line 139. Why do results of experiments at 1300 °C have any relevance here?

Line 142. The statement relating the Ce⁴⁺/Ce³⁺ ratio to oxygen fugacity and temperature refers to an experimental study of the behaviour of Ce in magmas, based on experiments at 1300 to 1500 °C. I don't see that it has any relevance in the discussion of secondary minerals in granite.

Line 153. Secondary minerals form at subsolidus conditions, commonly as a result of hydrothermal alteration. The statement that secondary REE minerals formed during "granite solidification" stretches credulity. As the authors emphasise "not during "the weathering stage", I suspect they mean post-solidification. Two lines down there is reference to "the fluid source" by which I suspect they mean a hydrothermal fluid.

Line 155. Meaning unclear.

Line 168. Why should late stage magmas alter zircon? Has any evidence been reported for this phenomenon?

Line 172. The reader needs to be reminded that the evidence for an oxidising metasomatic fluid is the unusually low Ce content of the secondary REE minerals. Somewhere, it needs to be pointed out that Ce⁴⁺ species are very immobile because of precipitation of the low solubility of cerianite. Thus an oxidising metasomatic fluid would be incapable of transporting significant Ce⁴⁺.

Line 177. Instead of reporting absolute values of oxygen fugacity, which are meaningless to most readers, report the oxygen fugacity with reference to a mineral buffer like QFM or HM.

Line 186. What is meant by the term "normal oxygen fugacity". Surely oxygen fugacity will vary with environment, and there is no such thing as a "normal oxygen fugacity".

Line 187. Most researchers would agree that peralkaline magmas emplaced in continental rift environments are-REE enriched and that these magmas, particularly the nepheline syenites and carbonatites originate in the mantle. How do the authors then reach the conclusion that mantle magmas "do not contain high levels of REE"?

Line 201. Cl⁻ and CO₃²⁻ are not minor ligands in aqueous fluids.

Line 208-210. What is the difference between an oxidized supercritical fluid and an aqueous fluid? Aqueous fluids can be both oxidizing and supercritical!!

Re: "Nature Communications" Manuscript Number: NCOMMS-16-01906

"Origin of heavy rare earth mineralization in South China" by Xu et al.

Dear Editor

Thank you for your letter and for returning the above manuscript with comments of three reviewers. We revised the manuscript according to their suggestions. Coauthor Dr. Martin Smith checked the last revision. The following is our point-by-point response to the three reviewers.

Response to the Reviewer #1:

1. Line 0: Detailed datasets, complete descriptions of the methodology, and additional figures are included in the supplementary information section. Tables S2 (compositions of mineral in the granite) and S3 (major and trace compositions of granite and fluorapatite and secondary REE minerals) mix whole rock data and mineral chemistry. Symbols in Figure 4a are not decipherable (too small, and lack contrast). Are color figures possible?

Re: The datasets are listed in the Supplementary information. The major Methods are shown in the context and additional analyses in the Supplementary information. The table tiles are revised. The Figure 4a is improved with large symbols and color lines, which is used to distinguish different minerals.

2. Line 40-42: The authors should cite additional recent papers describing REE IAC occurrences in Madagascar and the Southeastern United States (as suggested here).

Re: *These studies are now referenced.*

3. Line 68-69: The Author should cite a reference(s) for the statement describing bacterial fractionation of REE through organic complexation.

Re: *Reference now added.*

4. Line 100-102: The Authors may want to include a plot to illustrate the described relationship between bulk REE (concentrations) in the granite and Ce anomaly.

Re: *The relationship between La and Y contents and Ce anomaly of the granites are listed in Fig. 1. Both of them show an increase with increasingly negative Ce anomaly. The La and Y represent light and heavy REE, respectively, so bulk REE composition also increases with Ce depletion.*

5. Line 101: A recent paper by Foley and Ayuso (2015) describe some REE characteristics of granite hosts that may also be evident in REE IAC deposits of the Southeastern United States. They note the presence of REE tetrad behavior in many examples of granites that host IAC deposits. The data sets included in that paper show the presence of a negative Ce anomaly in granites of Xinxui HREE IAC and Zudong HREE IAC (including data from Bao and Zhou, 2008). It would be useful to add a plot showing

the REE pattern for the Zhaibei granites and describe whether similar tetrad behavior is present as this may support extreme fractionation and/or late magmatic - hydrothermal alteration, processes that are discussed as important in the petrogenesis of REE-enriched granite source rocks.

Re: *The data of Bao and Zhou (2008) are plotted in Fig. 1, which also show higher REE content with stronger negative Ce anomaly. This gives evidence related to the unknown Ce-poor minerals with both LREE and HREE enrichments in the primary granites.*

6. Line 109: These are remarkably large 'igneous' apatite crystals. Wang et al., 2015, describe the mineral suite for Zhaibei granite as including allanite, as well as monazite and apatite. It is noteworthy that igneous apatite is suggested to be the primary carrier of REE, rather than allanite. Late magmatic-hydrothermal alteration of igneous allanite can lead to formation of monazite, zoned apatite crystals + bastnäsite + calcite. Is there any evidence for REE-enriched overgrowths on apatite as has been described by Ishihara et al, 2008?

Re: *The sentence here says 'up to 1mm'. Not all apatites are this size, and 1 mm grain sizes are not that unusual in many systems. The fluoroapatite and monazite are primary REE carriers, not allanite in the granites. The metasomatic apatite, observed by Ishihara et al. (2008), only shows slightly REE enrichment in the mineral rim, and its major composition has little difference. Our work in this study has found new unknown REE*

mineral formation in the metasomatic fluorapatite. Therefore, the work of Ishihara et al. (2008) is not cited here.

7. Line 118: Change to 'metasomatized to form new REE minerals', or 'new phosphate and fluorcarbonate' (see Line 128).

Re: *This change has been made.*

8. Line 122-125: The paper could be improved by the addition of higher magnification photographs of the unknown mineral phases so that the described features could be better seen. This is an important addition because many of the arguments depend on the identification of the 'new' secondary REE minerals. Although the mineral data is likely insufficient at the present time to meet IMA standards for new mineral characterization, the paper raises an awareness of the need for further studies to identify and characterize unknown REE mineral phases in these systems.

Re: *New high resolution (4096*3536), magnification (* 10000 times) images of the unknown minerals have been added in Figure 2e, f.*

9. Line 139-143: Provide appropriate reference(s) to the experimental study described.

Re: *References now added.*

10. Line 141-145: Data for fluorapatite appears to be missing from Figure 4a. Symbols are difficult to distinguish.

Re: *The fluorapatite data are shown in Fig 4a. The figure has been improved to distinguish different mineral compositions.*

11. Comment on the impact of the breakdown of allanite (Wang et al, 2015) as a contributing mineral to the Nd budget in the bulk granite. Consider including Nd isotope data for the allanite.

Re: *Allanite is not a primary REE carrier and not common in the granites, so we analysed the Nd isotope of the primary fluorapatite and monazite.*

Response to the Reviewer #2:

1. I am surprised that no XRD data are reported, but the newly reported phases may be too rare to be detectable in rock powders (how abundant are they?). As mentioned by the authors, the initial Nd isotope ratios are somewhat difficult to determine given the changing parent-daughter element ratios during the envisaged metasomatism. The undertaken Nd isotope modeling appears sound. I would have preferred to see another isotope system showing similar results such as Rb-Sr, or Lu-Hf (note that Lu is a HREE and this could have produced interesting results). The literature references on carbonate-rich magmas or carbonatites could be updated; the work by Woolley is a bit outdated. There is an increasing body of evidence that carbonate melts in Earth's mantle are actually not that enriched in incompatible elements including the REE (see Foley et al., 2009, *Lithos*). This enrichment must also happen via zone refining and fractionation likely in the crust. For exceptionally REE enriched carbonatites and carbonate-bearing ultramafic silicate rocks (all

mantle-derived) see tables in Tappe et al. (2006, Journal of Petrology).

Re: *It is very difficult to pick up the unknown REE minerals from the granite for XRD analysis. Relatively low concentrations of mineral phases have major implications for the REE budget of the granites. In this instance we analysed these REE minerals using in-situ laser Raman. Neodymium isotopes are a direct tool for tracing REE source. These REE minerals contain very low Rb, Sr and Hf contents, so their isotopes cannot be analysed by in-situ laser-ablation MC-ICP-MS. The references for carbonatites with high REE are updated. It is reported that primary carbonatite magmas deriving from mantle peridotites may not contain high REE contents, but the REE can be enriched by zone refining and fractionation (Tappe et al., 2006; Journal of Petrology). Importantly, carbonated oceanic crust-derived carbonate-rich fluids have high REE contents, which can form large REE deposits (see listed references in Hou et al., 2015, Scientific Report, and Xu et al., 2015, Lithos).*

2. The manuscript is generally well written, but it still contains a few minor grammatical errors and awkward phrasing that should be taken care of during the revisions. Figure 1 is missing explanations of the legend, even in the caption. The panels in Figure 3 are arranged in a very awkward way. Figure 4a is a bit messy; Figure 4b is simple but powerful.

Re: *Our coauthor of Dr. Martin Smith read and checked the last revision. The caption in the Fig. 1 is modified. The Figs. 3, 4a are improved.*

3. Line 32-35: I fail to see the logic of that statement; some rephrasing may help.

Re: *We have rephrased this sentence.*

4. Line 129: it must say "laser-ablation MC-ICP-MS"; laser alone is not doing anything.

Re: *This has been changed.*

5. Line 137: check grammar.

Re: *This has been changed.*

6. Line 139, 177, 187: when reference is made to oxygen fugacity, it should be stated what the reference buffer is, i.e. FMQ or NNO or HM?

Re: *This is now done.*

7. Line 149-152: why not collecting extra Rb-Sr or Lu-Hf isotope data? This would be comforting for the rather big conclusions drawn.

Re: *The REE minerals contain very low Rb, Sr and Hf, which cannot be analysed by in-situ laser-ablation MC-ICP-MS for their isotopes.*

8. Line 157: check grammar.

Re: *This has been changed.*

9. Line 188: Not all subducted material is high in epsilon Nd; i.e. think of marine sediments in proximity to cratons.

Re: *Carbonate-rich liquids derived from melting of carbonated oceanic crust generally contain high epsilon Nd (see Xu et al., 2014, Geochim. Cosmochim. Acta), although some cases show relatively low epsilon Nd*

(most have positive epsilon Nd value). Note that the unknown REE minerals are characterized by formation under high oxygen fugacity conditions and have high epsilon Nd, further indicating a subducted fluid contribution.

10. Line 191-192: unclear; what has the subducted slab to do with the subcontinental mantle? Or do you mean asthenosphere or convecting mantle?.

Re: *The sentence is rewritten.*

11. Line 198: check grammar.

Re: *This has been changed.*

12. Line 217: Carbonatite references could be updated.

Re: *References have been updated.*

13. Line 226: Awkward. A magma cannot be metasomatized, but a rock or crystal mush can.

Re: *Now clarified to refer to sub-solidus conditions.*

14. Line 234: It is a deposit due to weathering processes, but not a "weathered deposit", correct?

Re: *Now corrected.*

Response to the Reviewer #3:

1. The term "superlarge" is inappropriate in the title.

Re: *Now changed.*

2. Line 19. What about the other LREE, e.g., La and Nd. The statement here might apply to Ce because it is fairly immobile in the 4+ state but would not apply to the other LREE.

Re: *We have clarified this statement.*

3. Line 23-25. The conclusion of this sentence does not follow from the preceding sentence. What is the basis for proposing that "REE-rich fluids.....metasomatised the granites during solidification"?

Re: *We have clarified this statement to make it clear. We are discussing subsolidus alteration.*

4. Line 51. For greater accuracy replace "solubility" with "REE" and "mobility" with "stability".

Re: *We have made these changes.*

5. Line 55. What about fluorcarbonate minerals? They are referred to on line 104.

Re: *We have clarified this statement.*

6. Line 53. Delete "role of".

Re: *This has been changed.*

7. Line 69. Supply a reference to support this statement.

Re: *Reference is now supplied.*

8. Line 74. Why would it not be sufficient to determine the compositions of the granites? Why is it necessary to know the composition of the corresponding magmas?

Re: *This statement has been clarified.*

9. Line 75. Why is Ce emphasized? Presumably concentrations of the other LREE are also low.

Re: *This is first report of REE-phosphates and carbonates with strongly depletion in Ce relative to La and Pr, which is an indicator of high oxidation fugacity.*

10. Line 92. The statement "The chondrite normalized (CN) REE patterns....." needs to be linked to a figure illustrating the profiles. Line 98. Ditto.

Re: *The sentences are re-written. The relationship between REE ratios and profile depth can directly find REE variety with increasing depth in Supplementary Fig. 1.*

11. Line 100. This statement, which is linked to Figure 1, can only apply to La and Y not REE generally.

Re: *The La and Y represent LREE and HREE, respectively, and both of the compositions show consistent increase with increasingly negative Ce anomaly. So this feature can be applied to total REE.*

12. Line 104. The IMA accepted name for sphene is titanite. REE minerals need to be given a suffix indicating the dominant REE. Thus, for example, monazite in Table S2 should be monazite-(Ce) Correct Table S2 accordingly. Line 105. See preceding comment about REE mineral names. Correct here and elsewhere.

Re: *Mineral nomenclature has been clarified.*

13. Line 135. How abundant are these unidentified REE minerals relative to xenotime-(Y), which is another potential source of HREE?

Re: *This statement has been clarified.*

14. Line 138. Explain the statement "This suggests that these minerals formed under high oxidation conditions".

Re: *We have added detail to this section to clarify the statement.*

15. Line 139. Why do results of experiments at 1300 °C have any relevance here? Line 142. The statement relating the Ce^{4+}/Ce^{3+} ratio to oxygen fugacity and temperature refers to an experimental study of the behaviour of Ce in magmas, based on experiments at 1300 to 1500 °C. I don't see that it has any relevance in the discussion of secondary minerals in granite.

Re: *This is a reference to one of the few available experimental studies on Ce oxidation in magmas, to illustrate the necessity for high oxygen fugacity. Moreover, the $\log(Ce^{4+}/Ce^{3+})$, and consequently $\log f_{O_2}$, have a linear dependence on $1/T$ (Burnham and Berry, 2014, Chemical Geology). This implies that the oxygen fugacity will be higher at the lower temperature.*

16. Line 153. Secondary minerals form at subsolidus conditions, commonly as a result of hydrothermal alteration. The statement that secondary REE minerals formed during "granite solidification" stretches credulity. As the authors emphasise "not during "the weathering stage", I suspect they mean post-solidification. Two lines down there is reference to "the fluid

source" by which I suspect they mean a hydrothermal fluid.

Re: *This section has been clarified to make it clear that we are referring to sub-solidus alteration.*

17. Line 155. Meaning unclear.

Re: *We have expanded this sentence to make it clear that it refers to model isotopic evolution trends.*

18. Line 168. Why should late stage magmas alter zircon? Has any evidence been reported for this phenomenon?

Re: *We have clarified this statement with reference to the available literature.*

19. Line 172. The reader needs to be reminded that the evidence for an oxidising metasomatic fluid is the unusually low Ce content of the secondary REE minerals. Somewhere, it needs to be pointed out that Ce⁴⁺ species are very immobile because of precipitation of the low solubility of cerianite. Thus an oxidising metasomatic fluid would be incapable of transporting significant Ce⁴⁺.

Re: *This point has been clarified.*

20. Line 177. Instead of reporting absolute values of oxygen fugacity, which are meaningless to most readers, report the oxygen fugacity with reference to a mineral buffer like QFM or HM.

Re: *We now make reference to common mineral buffers for oxygen fugacity.*

21. Line 186. What is meant by the term "normal oxygen fugacity". Surely oxygen fugacity will vary with environment, and there is no such thing as a

"normal oxygen fugacity".

Re: *We have removed this comment and replaced with a more specific sentence.*

22. Line 187. Most researchers would agree that peralkaline magmas emplaced in continental rift environments are-REE enriched and that these magmas, particularly the nepheline syenites and carbonatites originate in the mantle. How do the authors then reach the conclusion that mantle magmas "do not contain high levels of REE"?

Re: *We have removed this statement, and note that although nepheline syenites are very REE enriched at crustal levels, they achieve these concentrations through a range of magmatic evolution processes (fractional crystallization, immiscibility), and may not show the same levels of enrichment in primary magmas.*

23. Line 201. Cl⁻ and CO₃²⁻ are not minor ligands in aqueous fluids.

Re: *We have corrected this statement.*

24. Line 208-210. What is the difference between an oxidized supercritical fluid and an aqueous fluid? Aqueous fluids can be both oxidizing and supercritical.

Re: *We have modified this statement to make it unambiguous.*

Reviewers' Comments:

Reviewer #1 (Remarks to the Author)

This paper contributes to the growing body of knowledge on specific attributes of Chinese REE ion adsorption clay deposits and their possible genetic implications. As such, the Author's work may help focus exploration efforts in similar terranes outside of China.

The manuscript is much improved. The revised version of this manuscript addresses most of the major comments in my original review; only my comment on the implications of REE tetrad behavior in these igneous suites was left unaddressed. It is an ancillary point, as the focus here on Ce depletion in minerals as an indicator of high oxidation state and resulting metasomatism.

Reviewer #2 (Remarks to the Author)

Dear Nature Communications editors,

I was surprised how quickly the authors resubmitted their revised manuscript version, and after having looked through the ms and the rebuttal letter I arrive at the following conclusion. Cosmetic changes have been done to improve style and language. However, the scientific content was somewhat neglected. I had specifically asked to back up the conclusion with a second isotope system, which has not been done. Instead, low concentrations of Sr and Hf are provided as an excuse. There exist other methods than just LA-MC-ICPMS by which we routinely measure Sr and Hf isotopes (e.g., TIMS, solution MC-ICP-MS), and one will need to employ wet-chemical separation of elements in lower abundances. I also do not see where and how the literature has been updated. Figures are still too complex etc.

In conclusion, I think the authors have not taken the reviewer's comments all that serious, which is disappointing.

Kindly,

S. Tappe

Reviewer #3 (Remarks to the Author)

General Comments

This manuscript is very much improved from the original. Unfortunately, I have an important concern that I feel needs to be satisfactorily addressed before this manuscript can be accepted for publication, and this relates to the interpretation of the three unknown HREE minerals. The authors conclude on the basis of textures illustrated in Figure 2 that these minerals are the products of subsolidus metasomatism and not weathering. In my view, however, these textures and particularly that in Figure 2e do not allow this conclusion to be drawn. The textures could easily have resulted from weathering, i.e., interaction of the apatite with a fluid (meteoric water) at ambient rather than elevated temperature. Most significantly, only the three unknown HREE secondary minerals display negative Ce anomalies. The secondary monazite-(Ce) and Ca-REE fluorocarbonates show no such anomalies (Fig. 4a). Thus, if the unknown HREE phases are metasomatic, then it follows that there must have been two high temperature metasomatic events. The alternative is that the HREE phases formed during weathering, which may explain why they are unknown; minerals forming at low temperature commonly have complex stoichiometry that is difficult to interpret. Formation of these minerals during weathering would also satisfactorily explain the negative Ce anomalies; the meteoric fluids responsible for weathering would have been highly oxidizing. I apologise for not recognizing this issue in my earlier review. My remaining

comments are keyed to line numbers in the manuscript.

Comments keyed to the text

Lines 121-122. Replace the word "metasomatised" with altered. The latter word is descriptive, whereas "metasomatised" has a genetic connotation. Leave interpretation of the process until later.

Lines 144-149. As I noted in my previous review of this manuscript, reports of the oxygen fugacity (~ 11) and $Ce^{4+}/\Sigma Ce$ ratio (0.4) for the FMQ buffer at 1300 °C have no place in this manuscript. The secondary REE-phosphates to which this information is applied formed at some temperature below 750 °C, the likely solidus temperature of the host granite. Indeed, it is possible that they formed during weathering at near ambient temperature. Even if the data can be extrapolated to lower temperature (not attempted), there is no point in doing so, without knowing the temperature of formation of these secondary minerals. It is sufficient to conclude that the likely reason for the depletion in Ce relative to La and Pr is that the Ce was in the 4+ valence state, and then go on to point out that the anomalous behaviour was due to a combination of the difference in charge (mentioned in the manuscript) and the much smaller ionic radius of Ce^{4+} relative to Ce^{3+} (not mentioned in the manuscript); the ionic radius of Ce^{4+} is very similar to that of the HREE, Yb^{3+} .

Lines 158-159. I do not see that Figures. 2c, d and particularly Fig.2e (not mentioned) provide evidence that the secondary minerals did not form during weathering.

Line 174-176. Alteration of the zircon to xenotime-(Y) by late stage magmas seems unlikely and is not mentioned by the two references cited. They refer only to the alteration of zircon by hydrothermal fluids.

Fig. 4a. Primary and secondary monazite-(Ce) should be shown separately in Figure 4a.

Response to the Reviewer #1:

This paper contributes to the growing body of knowledge on specific attributes of Chinese REE ion adsorption clay deposits and their possible genetic implications. As such, the Author's work may help focus exploration efforts in similar terranes outside of China. The manuscript is much improved. The revised version of this manuscript addresses most of the major comments in my original review; only my comment on the implications of REE tetrad behavior in these igneous suites was left unaddressed. It is an ancillary point, as the focus here on Ce depletion in minerals as an indicator of high oxidation state and resulting metasomatism.

Re: We are particularly grateful to the referee for his support.

Response to the Reviewer #2:

I was surprised how quickly the authors resubmitted their revised manuscript version, and after having looked through the ms and the rebuttal letter I arrive at the following conclusion. Cosmetic changes have been done to improve style and language. However, the scientific content was somewhat neglected. I had specifically asked to back up the conclusion with a second isotope system, which has not been done. Instead, low concentrations of Sr and Hf are provided as an excuse. There exist other methods than just LA-MC-ICPMS by which we routinely measure Sr and Hf isotopes (e.g., TIMS, solution MC-ICP-MS), and one will need to employ wet-chemical separation of elements in lower abundances. I also do not see where and how the literature has been updated. Figures are still too complex etc.

Re: We crushed many granite samples to pick out REE minerals, but cannot obtain large amounts of pure unknown HREE-rich minerals of REE-1, REE-2 and REE-3 for solution isotope analysis. These unknown REE-1 and REE-2 are mixed with a

significant number of grains of monazite-(Ce) and apatite, and the REE-3 mixed with REE-fluorocarbonates. All of them contain similarly low Sr and Hf contents. So the solution isotopic analyses of mixed REE minerals cannot give credible evidence to explain HREE mineralization.

In the revision, we provide new isotope evidence from the results of Sr isotope analyses of Sr-rich calcite by in-situ LA-MC-ICPMS, and Sr, Nd isotopes of fresh feldspar by solution MC-ICPMS. The Sr-rich calcite (SrO ~ 1.8 wt%) in the granites is very rare, and similar to calcite of a typical carbonatitic origin, which may derive from subducted fluids metasomatizing the granites. Given the extremely low $^{87}\text{Rb}/^{86}\text{Sr}$ ratio (<0.002) of the calcites, their measured $^{87}\text{Sr}/^{86}\text{Sr}$ ratios are considered to approximate the initial isotope ratio. The calcites have lower initial Sr isotope ratios (0.7054-0.7061) than the primary feldspars (0.7088-0.7241) in the granites. This is consistent with the difference in Nd isotopic compositions between Ce-poor, HREE-rich minerals and primary LREE-rich minerals. The former has higher $\epsilon\text{Nd}(t)$ value. We would argue this indicates the fluids derived from subducted slab metasomatized the granites and formed HREE-rich minerals.

The references have been updated, and Figures have been modified and simplified.

Response to the Reviewer #3:

1. This manuscript is very much improved from the original. Unfortunately, I have an important concern that I feel needs to be satisfactorily addressed before this manuscript can be accepted for publication, and this relates to the interpretation of the three unknown HREE minerals. The authors conclude on the basis of textures illustrated in Figure 2 that these minerals are the products of subsolidus

metasomatism and not weathering. In my view, however, these textures and particularly that in Figure 2e do not allow this conclusion to be drawn. The textures could easily have resulted from weathering, i.e., interaction of the apatite with a fluid (meteoric water) at ambient rather than elevated temperature. Most significantly, only the three unknown HREE secondary minerals display negative Ce anomalies. The secondary monazite-(Ce) and Ca-REE fluorocarbonates show no such anomalies (Fig. 4a). Thus, if the unknown HREE phases are metasomatic, then it follows that there must have been two high temperature metasomatic events. The alternative is that the HREE phases formed during weathering, which may explain why they are unknown; minerals forming at low temperature commonly have complex stoichiometry that is difficult to interpret. Formation of these minerals during weathering would also satisfactorily explain the negative Ce anomalies; the meteoric fluids responsible for weathering would have been highly oxidizing. I apologise for not recognizing this issue in my earlier review. My remaining comments are keyed to line numbers in the manuscript.

Re: Figure 2e and f are high magnification images of unknown HREE-rich minerals. The black background of feldspar and biotite are fresh and not well shown because we used high contrast to highlight the HREE-rich mineral features. We have rewritten the description of monazite-(Ce) and REE-fluorocarbonates, some of which were formed by hydrothermal fluid alteration of LREE-rich minerals (possible mosandrite; Supplementary Fig. 2). It is normal in most granitoids that early LREE minerals in the rocks were altered to form new LREE minerals by granite-exsolved fluids. This is why we cannot find Ce-poor, HREE-rich minerals.

The textures are clearly metasomatic. This only means formation by chemical addition to the rock, and should carry no implication of hydrothermal or weathering

conditions. The Ce anomaly indicates oxidising conditions, but these are not exclusively indicative of weathering. Zircon commonly hosts Ce^{4+} in oxidised magmatic systems (e.g. Hoskin, 2005, Trace-element composition of hydrothermal zircon and the alteration of Hadean zircon from the Jack Hills, Australia. *Geochimica et Cosmochimica Acta* 69, 637-648. Trail et al., 2012, Ce and Eu anomalies in zircon as proxies for the oxidation state of magmas. *Geochimica et Cosmochimica Acta* 97, 70-87). Equally, igneous rocks can develop negative Ce-anomalies either by loss of Ce^{4+} to an oxidised magmatic-aqueous phase (e.g. MacDonald et al., 2008, the roles of fractional crystallization, magma mixing, crystal mush remobilization and volatile-melt interactions in the genesis of a young basalt-peralkaline rhyolite suite, the greater Olkaria volcanic complex, Kenya rift valley, *Journal of Petrology* 49, 1515-1547) or through inclusion of subducted oceanic crust in the magma source (e.g. Shimizu et al., 1992, Ce and Nd isotope geochemistry on island arc volcanic rocks with negative Ce anomaly: existence of sources with concave REE patterns in the mantle beneath the Solomon and Bonin island arcs, *Contributions to Mineralogy and Petrology* 110, 242-252).

Circulating meteoric water could contain high oxygen, and possibly REE deriving from dissolution of early crystallizing REE minerals in the granites, which metasomatized the rocks to form HREE-rich minerals. But the metasomatic fluids would inherit the high Sr and low Nd isotopic compositions of the host granites. This is clearly not the case as the HREE-rich minerals have higher $\epsilon Nd(t)$ (0.9 ± 0.8) than the primary LREE-rich minerals (-11.5 ± 0.5), feldspar (-2 to -13) and whole rocks (-0.8 to -7). Rare Sr-rich calcite (~ 1.8 wt%) has been found in the granites, and is similar to calcite of a typical carbonatitic origin. In the revision, the Sr-rich calcite was analysed by in-situ LA-MC-ICPMS, and contains lower initial Sr isotope (0.7054-

0.7061) than the primary feldspar (0.7088-0.7224) and whole rocks (~0.7110). There must therefore be a depleted source contributing to the granite and deposit REE budget, and known sources in the weathering environment cannot provide these. This is critically important for the origin of the world's currently most significant HREE resources.

2. Lines 121-122. Replace the word "metasomatised" with altered. The latter word is descriptive, whereas "metasomatised" has a genetic connotation. Leave interpretation of the process until later.

Re: It has been changed.

3. Lines 144-149. As I noted in my previous review of this manuscript, reports of the oxygen fugacity (~11) and $Ce^{4+}/\Sigma Ce$ ratio (0.4) for the FMQ buffer at 1300 C have no place in this manuscript. The secondary REE-phosphates to which this information is applied formed at some temperature below 750 C, the likely solidus temperature of the host granite. Indeed, it is possible that they formed during weathering at near ambient temperature. Even if the data can be extrapolated to lower temperature (not attempted), there is no point in doing so, without knowing the temperature of formation of these secondary minerals. It is sufficient to conclude that the likely reason for the depletion in Ce relative to La and Pr is that the Ce was in the 4+ valence state, and then go on to point out that the anomalous behaviour was due to a combination of the difference in charge (mentioned in the manuscript) and the much smaller ionic radius of Ce^{4+} relative to Ce^{3+} (not mentioned in the manuscript); the ionic radius of Ce^{4+} is very similar to that of the HREE, Yb^{3+} .

Re: This part has been rewritten and simplified. There are only two reference

reported on Ce anomaly at high temperature. We only emphasize that experiments show an oxygen fugacity (f_{O_2}) greater than the fayalite-magnetite-quartz (FMQ) buffer for a $Ce^{4+}/\Sigma Ce$ ratio < 1 . The $\log(Ce^{4+}/Ce^{3+})$, and consequently $\log f_{O_2}$, have a linear dependence on $1/T$. Whereas the titanite+magnetite+quartz assemblage in the granites indicates oxygen fugacities of ~ 1 (\square FMQ), which is too low to oxidize Ce^{3+} to Ce^{4+} .

4. Lines 158-159. I do not see that Figures. 2c, d and particularly Fig.2e (not mentioned) provide evidence that the secondary minerals did not form during weathering.

Re: The Figure 2e,f is high magnification images of unknown HREE-rich minerals. The black background of feldspar and biotite are fresh and not well shown because we use high contrast to highlight the HREE-rich mineral features.

5. Line 174-176. Alteration of the zircon to xenotime-(Y) by late stage magmas seems unlikely and is not mentioned by the two references cited. They refer only to the alteration of zircon by hydrothermal fluids.

Re: This part has been revised.

6. Fig. 4a. Primary and secondary monazite-(Ce) should be shown separately in Figure 4a.

Re: This part was re-written. The early LREE-rich minerals were altered by hydrothermal fluids to form minor monazite (Supplementary Fig. 2), which is normal in most granites. The hydrothermal fluids may exsolve from the granites. The primary monazite crystallizing from granite magmas, and decomposed products from LREE-

rich minerals have similar chemical composition, so they are not distinguished in Fig. 4a. This further indicates that the Ce-poor, HREE-rich minerals in this study do not form from granite-exsolved hydrothermal fluids.

Reviewers' Comments:

Reviewer #2 (Remarks to the Author)

A much improved version that I believe makes an important contribution to the field of HREE mineralizations. I recommend publication in Nature Communications.

Kindly,

Sebastian Tappe
Johannesburg

Reviewer #3 (Remarks to the Author)

General Comments

I have re-read the manuscript and the responses of the authors to the points raised in my previous review (second review). All except one of the points that I raised have been satisfactorily addressed. Unfortunately, in my view, the authors have not responded adequately to my criticism of their interpretation of the textures involving the "unknown REE minerals". In their response, they state "The textures are clearly metasomatic. This only means formation by chemical addition to the rock, and should carry no implication of hydrothermal or weathering conditions". This response would seem to indicate that they agree with my assertion that it is not possible to decide, based on textural relationships, whether the minerals were the products of subsolidus (hydrothermal) processes or weathering. In the revised manuscript, however, they state "However, mineralogical observations indicate they formed under subsolidus conditions, not during the weathering stage (Line 171-172)". This statement clearly rejects my assertion and is thus inconsistent with the response of the authors quoted above. In my second review, I also drew attention to the fact that, whereas the three unknown HREE minerals exhibit negative Ce anomalies, the secondary minerals (of presumed hydrothermal origin) exhibit positive anomalies. Although, I agree that weathering (oxidising conditions) is not the only way to obtain positive anomalies, the authors have not explained how subsolidus hydrothermal activity gave rise both to REE minerals (HREE and LREE) without Ce anomalies and unknown REE minerals with negative Ce anomalies. Finally, I note, that the authors agree that the initial Nd isotope composition of the unknown HREE minerals cannot be determined because the age of these minerals is unknown and the Sm/Nd ratio variable (Lines 167-170). For this reason they revert to the textural evidence that the unknown minerals formed under subsolidus conditions and not in a weathering environment. For these reasons, I remain unconvinced that the unknown HREE minerals are not products of weathering.

Comments Keyed to Text

Line 27. Replace "unknown" with unidentified.

Line 31 "show higher initial Nd isotope" . Vague. This sentence should refer to a particular isotope ratio.

Line 46-48. This statement needs to be supported by a reference.

Line 52. Add "the" at the beginning of the line. Check the rest of the manuscript for missing articles and the quality of English, generally.

Line 61. Replace " include" with "are". Delete "solution" .

Line 68. It does not follow that stronger complexation of HREE with fluoride and carbonate reduces HREE adsorption on clay minerals. Indeed, it could be argued that the higher concentration of the HREE in solution will promote their uptake through adsorption provided that the complexes have charges or functional groups favourable for adsorption - complexation will increase the

concentration of the HREE in solution thereby making them available for adsorption.

Line 69. Replace "absorption" with "adsorption".

Line 72. Supply a reference to support the statement "which occur as colloids".

Line 86. Replace "unknown" with "unidentified".

Line 134, Insert "respectively" after "wt%".

Line 199. Replace "deriving" with "derived".

Response Notes

Response to the Reviewer #2:

A much improved version that I believe makes an important contribution to the field of HREE mineralization. I recommend publication in Nature Communications.

Re: We are particularly grateful to the referee for his support.

Response to the Reviewer #3:

I have re-read the manuscript and the responses of the authors to the points raised in my previous review (second review). All except one of the points that I raised have been satisfactorily addressed. Unfortunately, in my view, the authors have not responded adequately to my criticism of their interpretation of the textures involving the "unknown REE minerals". In their response, they state "The textures are clearly metasomatic. This only means formation by chemical addition to the rock, and should carry no implication of hydrothermal or weathering conditions". This response would seem to indicate that they agree with my assertion that it is not possible to decide, based on textural relationships, whether the minerals were the products of subsolidus (hydrothermal) processes or weathering. In the revised manuscript, however, they state "However, mineralogical observations indicate they formed under subsolidus conditions, not during the weathering stage (Line 171-172)". This

statement clearly rejects my assertion and is thus inconsistent with the response of the authors quoted above. In my second review, I also drew attention to the fact that, whereas the three unknown HREE minerals exhibit negative Ce anomalies, the secondary minerals (of presumed hydrothermal origin) exhibit positive anomalies. Although, I agree that weathering (oxidising conditions) is not the only way to obtain positive anomalies, the authors have not explained how subsolidus hydrothermal activity gave rise both to REE minerals (HREE and LREE) without Ce anomalies and unknown REE minerals with negative Ce anomalies. Finally, I note, that the authors agree that the initial Nd isotope composition of the unknown HREE minerals cannot be determined because the age of these minerals is unknown and the Sm/Nd ratio variable (Lines 167-170. For this reason they revert to the textural evidence that the unknown minerals formed under subsolidus conditions and not in a weathering environment. For these reasons, I remain unconvinced that the unknown HREE minerals are not products of weathering.

Re: We understand the referee's point regarding weathering, as the textures of the HREE minerals clearly show they are secondary in origin. We deleted the statement in Line 171-172 "mineralogical observations indicate they formed under subsolidus conditions, not during the weathering stage". Although it is difficult to accurately obtain the initial Nd isotope composition of these secondary REE minerals because of unknown mineral age and variable Sm/Nd ratio in an open environment, their isotopic evolution trends for different

weathering periods show the metasomatic fluids have a more depleted source than the granites. Regardless of the age of secondary mineralization, Nd cannot be sourced from the host granitoids (our Figure 4b). The Sr isotope composition of Sr-rich calcite also supports this conclusion. Therefore, the isotopic compositions of calcite and HREE minerals mean that a simple granite weathering and supergene redeposition model cannot explain the HREE enrichment of the deposits. The HREE must be derived from a source external to the granite plutons. In the revision, leaching of HREE from weathering of an unexposed isotopically depleted source rock is considered, but such a source has not been identified in the study area. This leads us to propose an alternative model that hydrous, carbonate-rich fluids released from the subducted slab underneath Mesozoic South China metasomatized the granites.

We have now altered the text, however, to reflect both models, whilst noting that we cannot locate a viable REE source to explain the isotopic characteristics within the local weathered rock system.

We sincerely thank the reviewer comments. The incorrect writing has been revised, and new references are also listed. Coauthor Martin Smith read and checked the latest revision.

Reviewers' Comments:

Reviewer #2 (Remarks to the Author)

I re-read the manuscript and do not think that the concern of R3 regarding the potentially secondary nature of the HREE mineralization is a big problem. The radiogenic isotope data presented somewhat preclude a large influence of alteration and its long-term effects. I would be happy to see the study published.

Response Notes

Response to the Reviewer #2:

I re-read the manuscript and do not think that the concern of R3 regarding the potentially secondary nature of the HREE mineralization is a big problem. The radiogenic isotope data presented somewhat preclude a large influence of alteration and its long-term effects. I would be happy to see the study published.

Re: We are particularly grateful to the referee for his support.